# Interprofessional Collaboration and Diabetes Management in Primary Care: A Systematic Review and Meta-Analysis of Patient-Reported Outcomes

**DOI:** 10.3390/jpm12040643

**Published:** 2022-04-15

**Authors:** Mario Cesare Nurchis, Giorgio Sessa, Domenico Pascucci, Michele Sassano, Linda Lombi, Gianfranco Damiani

**Affiliations:** 1Università Cattolica del Sacro Cuore, 00168 Rome, Italy; nurchismario@gmail.com (M.C.N.); giorgio.sessa@gmail.com (G.S.); michelesassano92@gmail.com (M.S.); gianfranco.damiani@unicatt.it (G.D.); 2Fondazione Policlinico Universitario A. Gemelli IRCCS, 00168 Rome, Italy; 3Università di Bologna, 40126 Bologna, Italy; 4Università Cattolica del Sacro Cuore, 20123 Milan, Italy; linda.lombi@unicatt.it

**Keywords:** type 2 diabetes, patient-reported outcomes, primary care, interprofessional collaboration

## Abstract

The global spread of diabetes poses serious threats to public health requiring a patient-centered approach based both on interprofessional collaboration (IPC) given by the cooperation of several different health professionals, and patients’ perspective through the assessment of Patient-Reported Outcomes (PROs). The aim of the present study is to evaluate the impact of interprofessional collaboration interventions, for the management of type 2 diabetes in primary care settings, through PROs. A systematic review and meta-analysis was conducted querying the PubMed, Scopus and Embase databases. Out of the 1961 papers initially retrieved, 19 met the inclusion criteria. Interprofessional collaboration is significantly associated with an increase in both patient’s satisfaction (SMD 0.32 95% CI 0.05–0.59) and in the mental well-being component of the HRQoL (SMD 0.18; 95% CI 0.06–0.30), and there was also promising evidence supporting the association between an interprofessional approach and an increase in self-care and in generic and specific quality-of-life. No statistical differences were found, supporting the positive impact on IPC interventions on the physical component of the HRQoL, depression, emotional distress, and self-efficacy. In conclusion, the effect of IPC impacts positively on the few areas assessed by PROMs. Policymakers should promote the widespread adoption of a collaborative approach as well as to endorse an active engagement of patients across the whole process of care.

## 1. Background

Non communicable diseases have spread all over the world as a pandemic and are threatening healthcare system sustainability [1]. Among these, diabetes mellitus can be considered to be one of the biggest problems. In 2019 [2], 463 million people have diabetes mellitus worldwide, and this is expected to increase by 51% by 2045, with more than 90% of cases being type 2 diabetes mellitus (T2DM) [3]. Older adults have the highest prevalence of T2DM of any age group, and comorbidities are common, with more than 40% of individuals with T2DM having 3 or more comorbidities. T2DM combined with comorbidity is linked to higher mortality, poorer functionality and greater health service use [4,5,6,7] than T2DM alone [8]. Above all, a huge public health problem is represented by the co-presence of T2DM and psychological comorbidity, such as depression and/or diabetes-specific emotional distress [9], as it is associated with poorer treatment outcomes because both depression and diabetes distress have been shown to negatively affect type 2 diabetes through poor adherence and reduced self-care [10,11,12]. Furthermore, diabetes is one of the most relevant causes of economic loss, morbidity, and early mortality in the world [13]. The per capita cost burden associated with diabetes is two to four-fold greater than that of non-diabetic patients [14].

Chronic conditions follow an unpredictable trajectory over a prolonged period and commonly do not achieve a cure, so to provide effective care for people with chronic conditions and to facilitate the shift from a reactive health care system to one which proactively involves patients has been developed in the Chronic Care Model [15,16]. Extensive evidence has shown that the Chronic Care Model improves patient care and provides a framework for improved efficiency and outcomes [17,18], especially in a primary care setting [19]. The Chronic Care Model is based on “patient-centeredness”, requiring an interprofessional collaboration approach and taking into account the patients perspectives.

There is extensive evidence to support the benefits of team-based care [20]. Interprofessional collaborative (IPC) practice [21] can be defined as the cooperation of several health professionals, belonging to different health or social care professions, with the shared goal of increasing collaboration and patient-related care quality. Therefore, IPC is an additional key aspect of caring for patients with multiple chronic conditions [15,22]. Individuals affected by chronic conditions require continued interactions with the health care system and must make ongoing adjustments in daily life. In addition, many chronic diseases are preventable or modifiable through alterations of risky behaviors, lifestyle changes, and self-care practices [23]. Patients, their families, and their caregivers are called upon to manage difficult care and adopt significant behavior changes requiring “a complex and diverse set of skills” [24]. To provide effective care, it is important for health professionals to understand all these aspects in managing chronic illnesses and gain skills to apply these concepts in clinical practice.

As stated by the Chronic Care Model, the patients’ perspective could be elicited through the adoption of the “patient-reported outcomes” (PROs).

A PRO can be defined as “any report of the status of a patient’s health condition that comes directly from the patient, without interpretation of the patient’s response by a clinician or anyone else” [25]. The PROs can be measured though generic and/or disease-specific questionnaires [26,27], defined as PROMs, aimed at measuring functional status, health related quality of life, symptoms burden, personal experience of care, and health-related behaviors such as anxiety and depression.

The routine collection of e-PROs by healthcare providers in their clinical practice may help them to improve the quality of care through the monitoring of patient symptoms [28] to promote the identification of their unmet needs, and to foster a patient-centred approach by tailored treatment [29] to increase patient involvement and the individualization of patient care trajectories [30]. Assessing the reports coming directly from patients is integral to delivering high-value patient-centered care. The PROs have the potential to systematically incorporate patient input for improvement in both quality and cost of care.

Therefore, the aim of the present study is to assess the impact of interprofessional collaboration interventions for the management of type 2 diabetes in primary care settings, through PROs.

## 2. Materials and Methods

### 2.1. Study Design and Literature Search

A systematic review of the literature was carried out querying the following electronic databases: EMBASE, ISI Web of Knowledge, MEDLINE, from their inception to October 2021, without language restrictions. The Population, Intervention, Comparator, Outcome (PICO) model was used to frame the following guiding question of the systematic review: What is the impact of IPC intervention on PROs, in a primary care setting, among patients with type 2 diabetes? Each PICO domain corresponded to the following elements: (P) Patients with type 2 diabetes mellitus in primary care, (I) IPC, (C) usual care and (O) patients reported outcomes (PROs) [31]. To ensure the systematic review quality, the Preferred Reporting Items for Systematic Reviews and Meta-Analyses (PRISMA) checklist and flow-diagram was used [32]. Taking into account the search strategy conducted by Reeves et al. [21], the search string was constructed combining keywords such as “diabetes mellitus type 2”, “interprofessional collaboration”, “interprofessional team”, “patient reported outcome measures”, “patient reported outcomes”, “health related quality of life”, “primary health care”, “primary care” and their synonyms through Boolean operators “AND” and “OR” (Appendix A). Finally, additional studies were identified by “hand search” of references from articles included in the review (i.e., snowball searching).

### 2.2. Study Selection

Two investigators independently screened titles and abstracts of all records to identify potentially relevant publications. The inclusion criteria for this review were: randomized clinical trials, published in English or in Italian, assessing, through PROs, IPC interventions on DM2 management compared to usual care in the primary care setting.

Technically, IPC implies a set of different interventions usually taking place in healthcare settings as well as by specific tools such as interprofessional checklists and meetings, pathways, and forms [21,33]. Scientific evidence showed a conceptual skepticism regarding the wide number of terms referring to IPC given the great amount of manuscripts written by expert health professionals rather than scholars. However, as a whole, IPC’s main characteristic is the interactive effort and the support of professionals deemed to be of paramount importance to achieving the outcome implying high levels of communication, mutual planning, collective decisions and common responsibilities [21]. “Usual care” was defined as the care the targeted patient population would be expected to receive as part of the normal practice without explicitly stressing any degree of collaboration [34]. Articles were excluded if they did not meet the inclusion criteria or if they met at least one of the following exclusion criteria: not peer-reviewed studies, studies including students in the IPC teams, and studies reporting PROMs assessing any construct that was not present in other manuscripts. The evaluation of the eligibility criteria was performed independently by the two authors and, in case of divergence, a third researcher was consulted.

### 2.3. Quality Assessment

Two investigators independently assessed the quality of the studies using the National Institute of Health’s Quality Assessment of Controlled Intervention Studies [35]. If disagreements occurred, the final decision was reached by team consensus. The tool assesses 14 parameters for evaluating the internal validity of a study. For each item, the investigator could select “yes”, “no”, or “cannot determine/not reported/not applicable” [35]. A potential risk of bias was considered if the item was rated as “no” or “cannot determine/not reported/not applicable” were selected for the items by the reviewer. If the “yes” answers were ≥75% of the total, an article was considered to be of “good” quality; if they were <75% but ≥50%, an article was scored as “fair”; if they were <50%, the article was scored as “poor”.

### 2.4. Data Extraction and Data Analysis

Two reviewers independently performed data extraction and a standardized form was used to tabulate the following data: bibliographic details, country, intervention team, setting, intervention, population, PROMs, main results. Considering the variety of different PROMs investigating the same area, we grouped them in order to conduct a meta-analysis.

For each study, standardized mean differences (SMD) between compared groups were computed using Hedge’s g statistics [36]. Thus, pooled estimates were obtained using the Paule-Mandel random-effects model [37,38], and between-study heterogeneity was assessed using I^2^ statistics, which describes the percentage of variability in estimates across studies due to chance rather than sample error [39,40].

All meta-analyses were performed using statistical software STATA (version 14.0; College Station, TX, USA) and two-sided *p* values < 0.05 were considered statistically significant.

## 3. Results

### 3.1. Study Selection

The literature search resulted in 1961 studies. After eliminating duplicates, the research team reviewed a total of 1725 manuscript titles and abstracts. A total of 48 full articles were considered potentially relevant and were reviewed by two independent researchers. After full text examination, 29 of 48 articles were excluded as they did not fulfill the selection criteria. The remaining 19 studies [41,42,43,44,45,46,47,48,49,50,51,52,53,54,55,56,57,58,59] were included in the systematic review and studies were considered for the meta-analysis (Figure 1).

### 3.2. Characteristics of the Studies

The included studies were published between 1998 and 2020, of which three were from Canada [46,50,54] and the Netherlands [48,59], while two were from the USA [42,44], Brazil [45,57] and the UK [56,58]. Overall, 6273 patients were enrolled in the 19 studies (range: 29–507), nine of which enrolled fewer than 100 patients.

The majority of the studies, 18 out of 20, were targeted at improving the role of diabetes’ patients in self-management and modifying lifestyle behaviors. Following the definition of Ismail et al. [60], 11 studies included educational intervention [43,46,47,48,49,50,51,52,54,56,57], 11 studies were characterized by psychological intervention [42,43,44,45,47,50,53,54,55,58,59], and three were based on peer support programs [41,43,55]. Other kinds of intervention assessed were medication control and the retraining of health professionals. The vast majority of the interventions were provided in outpatient settings, while three interventions were delivered through the adoption of telemedicine [44,47] and one intervention occurred directly at the patient’s home [50].

In many studies, it was clearly reported that an empowerment approach [43,49,56,57], patient engagement [54,55], and/or community engagement [41,46,54] were adopted. On the basis of the intervention provision, the target was person-based in 10 studies [44,45,47,50,51,52,53,55,58,59] while in six it was either in groups [42,43,48,49,56,57], community-based [41,46], or both [54].

“Nurse” was the most represented job category available in the intervention team in 14 studies [41,42,44,46,47,48,49,50,51,53,54,56,58,59], followed by “dietician” in eight studies [45,46,48,49,51,54,55,56] and “primary care physician” in 10 papers [43,45,47,51,52,53,55,58,59]. "Psychologist" was present in the intervention team in only five studies [42,43,45,53,59].

Overall, five studies assessed the impact of IPC in diabetes patients who had any kind of psychological symptoms or emotional distress [42,44,50,54,59].

A summary of the characteristics of each study is reported in Appendix A.

### 3.3. Quality Assessment

A score of ten or greater was indicative of good methodological quality, nine to seven was fair and studies scoring below seven were deemed to be of poor quality. The overall methodological quality of all included studies (*n* = 19) is summarized in Appendix A. Thirteen studies [41,42,44,46,47,48,49,50,52,53,54,58,59] were deemed of good quality, while five [43,51,55,56,57] were rated to be of fair quality, showing a moderate risk of bias. Only one study was rated as poor [45]. The most frequent quality criteria met were the use of randomization and the power of study calculation. A number of items were rarely reported, including those regarding adherence to treatment or avoiding other interventions. The drop-out rate at the endpoint was 20% or lower in 16 studies [41,42,43,44,46,47,48,49,50,51,52,53,54,55,57,59].

### 3.4. Data Synthesis

Different PROMs investigating the same area were grouped into nine categories (i.e., health-related quality-of-life—physical; Health-related quality-of-life—mental; Depression; Emotional distress; Patient’s satisfaction; Self-efficacy; Self-care; Quality of life—generic; Quality of life—specific), as already highlighted in the scientific literature [61]. Figure 2 depicts the results of the meta-analysis.

#### 3.4.1. Health-Related Quality-of-Life—Physical

Seven [41,42,44,45,46,49,54] trials assessed the physical component of the health-related quality of life on generic PROMs. In particular, five [42,44,45,46,54] were SF-12 questionnaires while two [41,49] were SF-36 ones. After pooling these studies, no significant difference between intervention and the usual care was found: SMD 0.05 (95% CI −0.03, 0.14).

#### 3.4.2. Health-Related Quality-of-Life—Mental

This was evaluated in seven trials using the SF-12 questionnaire in five of them [42,44,45,46,54] and the SF-36 questionnaire in two [41,49]. After pooling the studies, statistical analysis supports a significant difference in favor of the IPC intervention team: SMD 0.18 (95% CI 0.06, 0.30).

#### 3.4.3. Depression

Nine studies [42,44,45,46,47,50,54,55,56] investigated the depression scores on itemized scales. Four [42,46,54,55] used the Center for Epidemiologic Studies Depression Tool (CES-D), two [44,45] used the Beck Depression Inventory (BDI), two used the [50,56] Hospital Anxiety and Depression Scale (HADS), and one [47] used the Major Depressive Syndrome (PHQ-9). The pooled analysis did not show any significant difference between intervention and usual care: SMD −0.19 (95% CI −0.40, 0.02).

#### 3.4.4. Emotional Distress

This was assessed in four studies [51,55,56,59] adopting the Problem Areas in Diabetes (PAID) questionnaire. After pooling these trials, there was no significant difference in either group: SMD 0.00 (95% CI −0.18, 0.19).

#### 3.4.5. Patient’s Satisfaction

Two studies [51,52] assessed the patient’s satisfaction using the Diabetes Satisfaction and Treatment Questionnaire (DTSQ). After analyzing these trials together, a significant difference was found between intervention and usual care: SMD 0.32 (95% CI 0.05, 0.59).

#### 3.4.6. Self-Efficacy

This was evaluated in six studies [46,47,50,53,54,55] adopting the Self-Efficacy for Managing Chronic Disease scale (SEMCD) in two [46,54] of them, the Diabetes Self-Efficacy Scale (DSES) in two [47,50], the Chinese diabetes management self-efficacy scale (CDMSES) in one [53], and the 20-item Diabetes Management Self-Efficacy Scale (DMSES) in the last one [55]. The pooled analysis highlighted no significant difference in either group: SMD 0.09 (95% CI −0.02, 0.19).

#### 3.4.7. Self-Care

Four trials [46,53,54,57] analyzed the self-care aspect. Three studies [46,53,54] implemented the Summary of Diabetes Self-Care Activities (SDSCA) scale while one [57] adopted the Self-care for type 2 diabetes (SLC). The statistical analysis did not show any significant difference between intervention and standard care: SMD 0.10 (95% CI −0.07, 0.28).

#### 3.4.8. Quality of Life—Generic

Four studies [43,47,48,56] reported the quality of life on generic self-report questionnaires. One [43] study used the Thai version of the World Health Organization Quality of Life—BREF, (WHOQOL-BREF) while one study [56] adopted the classic version of the same questionnaire.

Besides, one [54] study used the Assessment of Quality of Life (AQoL) Mark 2 instrument and one [48] study applied the three level EuroQol Five Dimension (EQ-5D-3L) scale. After pooling these trials, no statistical difference was found in either group: SMD 0.53 (95% CI −0.17, 1.23).

#### 3.4.9. Quality of Life—Specific

This was appraised in three studies [42,58,59] using the Diabetes version of the Ferrans and Powers Quality of Life Index (QLI) in one [42] of them, the Audit of Diabetes dependent quality of life (ADDQoL) in one [58], and the Diabetes Symptom Checklist—Revised (DSC-R) in the final one [59]. The metanalysis did not report any significant difference between the IPC intervention team and the standard alternative: SMD 0.14 (95% CI −0.03, 0.31).

## 4. Discussion

This study was intended to investigate, through the PROs, the role of interprofessional collaboration for the management of type 2 diabetes in primary care settings.

The present systematic review and meta-analysis pointed out that collaborative practice is significantly associated with an increase in both patient satisfaction and in the mental well-being component of the health-related quality-of-life.

There was also promising evidence supporting the association between the interprofessional approach and an increase in self-care and in generic and specific quality-of-life.

Disease-specific questionnaires are characterized by a set of questions aimed at investigating health changes related to a particular pathology, disability or intervention. These tools have a higher sensitivity, being able to intercept even small modifications in the analyzed disease. However, different from generic PROMs, specific questionnaires cannot be adopted to compare the health status among different conditions [62]. Hence, an overall evaluation should be based on the adoption of both the two types of questionnaires, generic and specific, which are to be considered complementary rather than in opposition for the assessment of the reported patients’ outcomes [63].

Moreover, there was a lack of evidence supporting the positive impact on IPC interventions on the physical component of the health-related quality-of-life, depression, emotional distress, and self-efficacy due to inconsistent findings. There are many possible reasons for this. One concern is related to the duration, complexity and intensity [64] of the intervention as well as the length of follow-up that may have been insufficient to see improvements [46]. Indeed, most of the studies lasted less than thirteen months. On the other hand, for some of the studies there is the possibility that the evaluation was premature and that patients had not been exposed to the intervention for long enough to detect any changes or the maximum change.

Moreover, the lack of consistent effect in some studies may in part be explained by the lower intensity [47] of intervention, especially in that study that based treatment on coaching techniques.

A more intensive telephone counseling intervention with more frequent calls, longer interaction, or longer duration of follow-up may lead to better outcomes. Also, the role of training of health professionals on coaching or educational intervention might be a reason for the results of some inconsistent findings, as most interventions had these characteristics. For example, motivational interviewing was originally developed for substance abuse, requiring a single behavioral change, whereas diabetes is a complex chronic illness that requires multiple behavioral changes [65], thus implying the lack of favorable effects on patient outcomes.

Another explanation to the lack of statistically significant findings lies in the good quality of the usual care approach for persons with diabetes in both the intervention and control groups, thereby avoiding that the training programme hardly added value [65].

In multicentric studies, it is also possible that the background and experience of healthcare providers of diabetes care also differed among the sites, which may have affected the findings. Another issue [66] that could influence the effect of the intervention is the Hawthorne effect [67]. On one hand, in an RCT the effect size could be underestimated, as both the intervention and control groups could improve their performance by virtue of participation in a study in which both groups were motivated [46] to implement an intervention to improve their performance. On the other hand, the effect could be overestimated in a controlled before-after study in which the control group provides the usual care and is not necessarily motivated to implement an intervention and is possibly not (completely) informed about the intervention and the purpose of this. The intervention group could improve their delivered care just because they participate in a study aimed at improving diabetes management.

For what concerns the physical aspect of the health-related quality-of-life, in the paper taken into the exam, only few interventions were focused on the promotion of the daily physical activities that represent an essential component of the questionnaire and, as highlighted by the evidence in the scientific literature [68], they are also fundamental for the proper management of type 2 diabetes.

The careful reading of the study findings allows for few key implications. The first suggests that IPC has the potential to ease healthcare processes, improve patient outcomes and care continuity and coordination as well as to reduce health costs in primary care [69].

Furthermore, given the fragmentation [70] of diabetes services characterizing health care systems and the high number of specialists involved, the integrated care model could represent a potential solution to obtaining a continuous multi-organizational assistance [70,71].

Lewis et al. [72] identified four different kinds of integration: organizational, functional, service and clinical. Given that IPC can be configured as both a service and clinical integration model, therefore it could be important to also focus on the organizational and functional integration.

The last implication regards the significance of endorsing patient-centeredness [73,74] for diabetes and other chronic conditions care, through the assessment of PROs and the integration of PROMs in clinical practice, by allowing a higher degree of patient involvement in the entire care process and the easing of the communication between health professionals and patients.

The findings of this systematic review and meta-analysis must be gauged in light of its strengths and weaknesses. Above all, the comprehensive and rigorous search strategy, the meticulous quality assessment and the methodological appropriateness in conducting the meta-analysis are strengths of the study. A limitation is represented by the merging of different PROMs, as illustrated by the lack of equal questionnaires, on the basis of the area investigated, thus leading to a potential increase in the heterogeneity. Nonetheless, we proceeded to standardize the reported estimates to allow a higher comparability during the meta-analysis process. Another caveat is the significant heterogeneity shown by some studies used for pooled analysis. However, it could be explained by clinical (i.e., type of intervention) diversity among the pooled studies. An additional limitation is the follow-up time of the selected studies, which may limit the applicability and validity of results and also may represent a hurdle in the proper identification of the long-term hazards.

Further research is needed to reach a broader consensus and to define a guideline on which PROMs should be adopted in diabetes management. Additional studies should define the intensity of the people-centered approach interventions, such as coaching or education, and which competencies/skills are needed for a productive interprofessional team in order to deliver an efficient and effective intervention.

## 5. Conclusions

In brief, this systematic review and meta-analysis brings a new and strong contribution to the literature debate on the impact of interprofessional collaboration interventions for the management of type 2 diabetes in primary care settings through PROs.

Currently, in a context characterized by an aging population, resource constraints and elevated health expenditures for chronic disease management, decision-makers should promote policies aimed to enhance the widespread adoption of a collaborative approach as well as to endorse the active engagement of patients across the whole continuum of care.

## Figures and Tables

**Figure 1 jpm-12-00643-f001:**
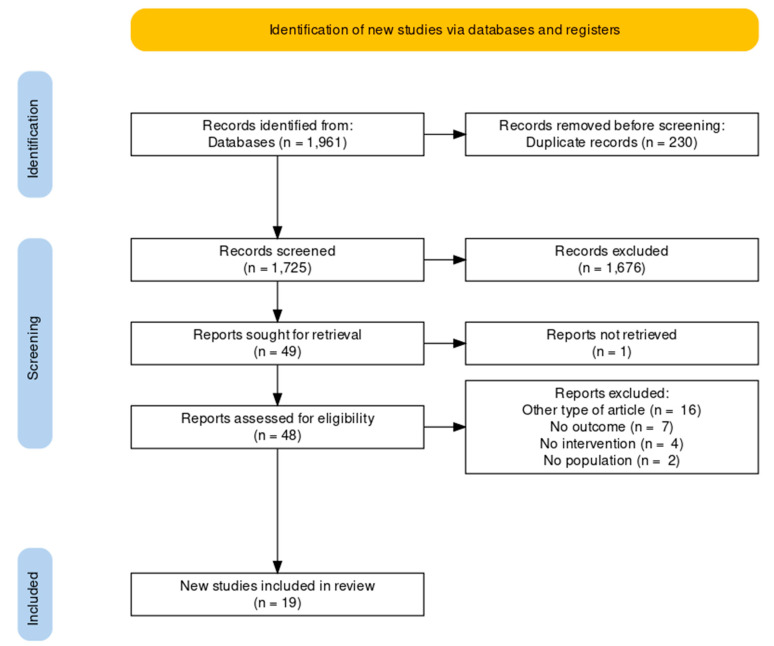
Flow diagram.

**Figure 2 jpm-12-00643-f002:**
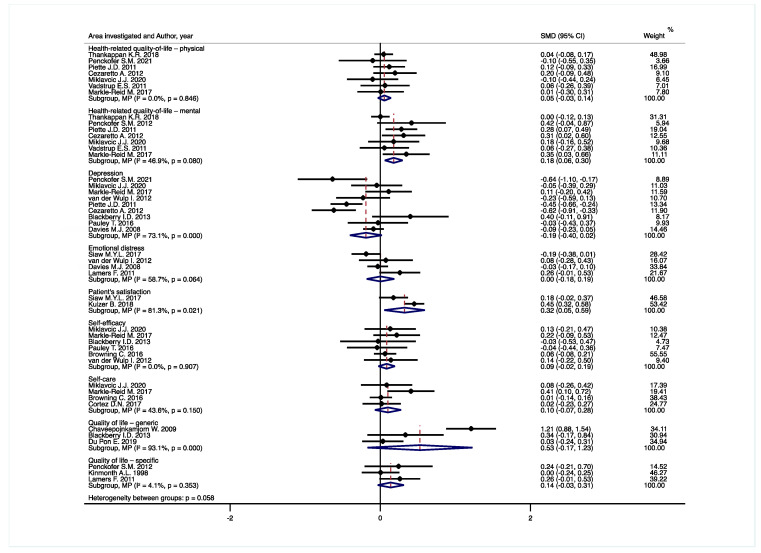
Findings of the pooled analysis of the included studies. Abbreviation: SMD, Standardized Mean Difference.

## Data Availability

The data presented in this study are available on request from the corresponding author.

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
