# Peer review of "Interprofessional Collaboration and Diabetes Management in Primary Care: A Systematic Review and Meta-Analysis of Patient-Reported Outcomes"

_jpm, 2022, doi:10.3390/jpm12040643_

Round 1

Reviewer 1 Report

This study has the appropriate methodology.
PICO has utility and increased its quality based on the meta-analysis.

The authors could discuss more about the follow-up in the selected studies, in addition to the duration, similarity or differences in the results at the measurement or evaluation cut-off points of the patients, what trends were observed during those follow-ups and perhaps factors at which that these variations were attributed during the follow-ups. In table S2 in which the 13 quality criteria that give clarity to the
results can be added.

Author, Year, Country

Source

Intervention team

Setting

Type of Interventions

Target of Intervention

Population

Follow-up

PROMs Analyzed

Results

Thankappan K.R. 2018, India

PLoS Medicine

 - Nurse with a PhD in public health
 - Medical Social Worker
 - Trained lay people

Outpatient clinics 

- Community engagement
- Peer support

Community-based

- 500 intervention group
- 507 control group

24 months

- SF-36 converted in Short Form 6 Dimension (SF-6D)

A community-based program didn’t result in a nonsignificant reduction in diabetes incidence. However, there were significant improvements in some cardiovascular risk factors and physical functioning score of the HRQoL scale.

Penckofer S.M. 2012, USA

Annals of Behavioral Medicine

- Nurse
- Nurse trained by psychologist 
- Psychologist

Outpatient clinics

- Psychological (psychoeducational)

Group-based

- 38 intervention group
- 36 control group

6 months

- SF-12
- CES-D
- QLI Diabetes version

SWEEP program provides evidence that group-therapy was more effective than usual care for treating depressed women with type 2 diabetes.

Chaveepojnkamjorn W. 2009, Thailand

Southeast Asian Journal of Tropical Medicine and Public Health

- Psychologist
- Diabetologist

Community Health Centers

- Educational (self-management)
- Empowerment-based
- Psychological (health belief model)
- Peer support

Group-based

- 80 intervention group
- 84 control group

6 months

-WHOQOL-BREF (THAI version)

This program, focused on enhancement of experience sharing among group members and participation in problem-solving, shows it is effective for improving perceived quality of life.

Piette J.D. 2011, USA

Medical Care

 - Nurses with psychiatric and primary care training and experience, trained in CBT
- Experienced CBT supervisor and trainer

Primary Care Clinics

- Psychological (Cognitive-Behavioural Therapy)
- Telemedicine

Person-based

- 172 intervention group
- 177 control group

12 months

- SF-12
- BDI

This program of telephone delivered CBT combined with a pedometer-based walking program did not improve A1c values but significantly decreased patients’ blood pressure, increased physical activity, and decreased depressive symptoms. The intervention also improved patients’ functioning and quality of life.

Cezaretto A 2012, Brazil

Quality of Life Research

-Endocrinologist
- Psychologist
- Nutritionist
- Physical Educator

Primary Care Clinics

- Psychological (psychoeducational)

Person-based

- 97 intervention group
- 90 control group

9 months

- SF-36
- BDI

An intensive intervention on lifestyle with interdisciplinary approach for individuals at risk for type 2DM induced greater improvements in QoL than a traditional one, in parallel to better benefits on cardiometabolic profile.

Miklavcic J.J. 2020, Canada

BMC Geriatrics

- Nurse
- Dietitian
- Program Coordinator

Primary Care Clinics

- Educational (self-management)
- Community engagement

Community-based

- 70 intervention group
- 62 control group

6 months

- SF-12
- CES-D
- SEM-CD
- SDSCA

This pragmatic trial of a self-management intervention for older adults with T2DM and multimorbidity demonstrated inconclusive results for improving QoL.

Blackberry I.D. 2013, Australia

BMJ Online

- Practice nurses
- General Practitioner

General practices

- Psychological (coaching)
- Telemedicine
- Educational (self-management)

Person-based

- 30 intervention group
- 29 control group

18 months

- AQoL
- PHQ-9
- DSES

A telephone coaching by existing generalist practice nurses without prescribing rights found no evidence that was effective compared with usual primary care, either in reaching treatment targets or achieving more intensive treatment.

Du Pon E. 2019, Netherlands

BMC Endocrine Disorders

- Practice Nurses 
- Dieticians specialized in diabetes care

General practices

- Educational (self-management)

Group-based

- 101 intervention group
- 100 control group

12 months

- EQ-5D-3L

PRISMA did not improve self-reported outcomes in patients with type 2 diabetes treated in primary care.It was not possible to make a statement about the clinical effects.

Vadstrup E.S. 2011, Denmark

Health Qual Life Outcomes

- Nurse
- Physiotherapist 
- Podiatrist
- Dietician

Outpatient Clinics and
General Practices

- Educational (self-management)
- Empowerment-based approach

Group-based

- 70 intervention group
- 73 control group

6 months

- SF-36

After 6 months this study suggests that a group-based rehabilitation programme is not superior to an individual counselling programme in changing patients' HRQOL and self-rated health.

Pauley T. 2016, Canada

Home Health Care Services Quarterly

- Nurse
- Personal Support Worker

Home 

- Psychological (coaching)
- Educational (self-management)

Person-based

- 47 intervention group
- 47 control group

1 month

- DSES
- HADS

A PSW-led coaching intervention to improve diabetes self-efficacy shows no differences compared to nurse-led traditional standard of care. However, the results do demonstrate it may be sufficient to improve depression. Furthermore all subjects demonstrated significant improvements in self-efficacy measures.

Siaw M.Y.L. 2017, Singapore

Journal of Clinical Pharmacy and Therapeutics

- Physicians
- Diabetes Nurse educators
- Dietitians
- Clinical pharmacists

Outpatient clinics

- Medication control
- Educational (self-management)

Person-based

- 214 intervention group
- 197 control group

6 months

- DTSQ
- PAID

After 6 months a multidisciplinary collaborative care for Asian diabetic patients there was an increased QoL and satisfaction towards diabetes care, lightened the physicians’ workload and cost saving. This study shows the effectiveness of this approach through an improvement of positive clinical, humanistic and economic outcomes.

Kulzer B. 2018, Germany

Diabetes Research and Clinical Practice

- Physicians received training based on a structured curriculum
- Medical staff

General practices

- Educational (self-management) 

Person-based

- 440 intervention group
- 467 control group

12 months

- DTSQ

The iPDM process improved the use of diagnostic data leading to better glycemic control, more timely treatment adjustments (indicating reduced clinical inertia), and increased patient adherence and treatment satisfaction among patients and physicians.

Browning C. 2016, China

BMJ Open

- Community Doctors trained in coaching
- Community Nurses trained in coaching
- Community psychologists trained in coaching

Community health Centers

- Psychological (health coaching - motivational intervention)

Person-based

- 372 intervention group
- 339 control group

12 months

- SDSCA
- CDMSES

In this study, although a differential treatment effect was not observed for HbA1c, numerous outcomes (including HbA1c) improved in both groups, supporting the establishment of regular, free clinical health checks for people with T2DM in Chinese CHSs.

Markle-Reid M. 2017, Canada

Journal of the American Geriatrics Society

- Nurse
- Dietitian
- Program Coordinator (PC) from a community partner
- Peer volunteers

Primary Care Clinics

- Educational (self-management) 
- Psychological (health coaching - motivational intervention) 
- Individual and community engagement

Community-based and Group-based

- 80 intervention group
- 79 control group

6 months

- SF‐12
- SDSCA
- CES‐D
- SEM-CD

This study provide evidence that participation in a 6-month community-based intervention improved quality of life and self-management and reduced depressive symptoms in older adults with T2DM and comorbidity without increasing total healthcare costs.

van der Wulp I. 2012, Netherlands

Diabetic Medicine

- Expert patients trained in motivational interviewing
- General Practitioner
- Dieticians

General practices

- Peer support
- Psychological (self-management coaching)
- Patient engagement

Person-based

- 59 intervention group
- 60 control group

6 months

- CES-D
- DMSES
- PAID

A peer-led self-management coaching programme for recently diagnosed patients with Type 2 diabetes improved self-efficacy of patients experiencing low self-efficacy shortly after diagnosis

Davies M.J. 2008, UK

BMJ

- Dieticians
- Practice Nurses
- Nurse specialists

General practices

- Educational
- Patient empowerment

Group-based

- 437 intervention group
- 387 control group

12 months

- WHOQOL-BREF
- HADS
- PAID

A structured group education programme for patients with newly diagnosed type 2 diabetes resulted in greater improvements in weight loss and smoking cessation and positive improvements in beliefs about illness but no difference in haemoglobin A(1c) levels up to 12 months after diagnosis.

Cortez D.N. 2017, Brazil

BMC Public Health

- Lead researcher as a facilitator and an instigator of discussions
- Research assistant

Primary Care Clinics

- Educational
- Patient empowerment

Group-based

- 127 intervention group
- 111 control group

12 months

- SLC

The empowerment program based on individualized goals was effective in improving self-care practices and metabolic control of type 2 diabetes in Brazilian users

Kinmonth A.L. 1998, UK

BMJ

- Nurse
- General Practitioner

General practices

- Psychological

Person-based

- 142 intervention group
- 108 control group

12 months

 - ADDQoL

A training programme in patient centred care for practitioners led to patients with newly diagnosed diabetes reporting better communication with doctors, greater wellbeing, and greater treatment satisfaction at one year, without loss of glycaemic control.

Lamers F. 2011, Netherlands

Journal of Advanced Nursing

- Nurses
- Psychiatrist
- General Practitioner
- Psychologist

General practices

- Psychological (Cognitive Behavioral Therapy)

Person-based

- 105 intervention group
- 103 control group

9 months

- DSC-R
- PAID

The nurse-administered intervention had limited effects on diabetes-specific quality of life.

Abbreviations: SF-36, short form health survey 36; SF-12, short form health survey 12; CES-D, The Center for Epidemiologic Studies Depression Tool; QLI Diabetes version, Diabetes version of the Ferrans and Powers Quality of Life Index; WHOQOL-BREF (THAI version), Thai abbreviated version of World Health Organization Quality of Life; BDI, Beck Depression Inventory; SEM-CD, Self-Efficacy for Managing Chronic Disease scale; SDSCA, Summary of Diabetes Self-Care Activities scale; AQoL, Assessment of Quality of Life Mark 2 instrument; PHQ-9, Major depressive syndrome; DMSES, 20-item Diabetes Management Self-Efficacy Scale; DSES, Diabetes Self-Efficacy Scale; EQ-5D-3L, EuroQol Five Dimensions scale, HADS, Hospital Anxiety and Depression Scale; DTSQ, Diabetes Satisfaction and Treatment Questionnaire; PAID, Problem Areas in Diabetes  questionnaire; WHOQOL-BREF, short version of the World Health Organization Quality of Life instrument; SLC, Self-care for type 2 diabetes; ADDQoL, audit of diabetes dependent quality of life; DSC-R, DiabetesSymptom Checklist – Revised.

Table S2. Results of quality assessment process of Controlled Intervention studies.

Author, Year, Country

1

2

3

4

5

6

7

8

9

10

11

12

13

14

Quality Rating

Thankappan K.R. 2018, India

yes

yes

yes

yes

no

yes

yes

yes

yes

yes

yes

yes

yes

yes

good

Penckofer S.M. 2012, USA

yes

yes

yes

no

no

yes

yes

yes

yes

NR

yes

yes

yes

yes

good

Chaveepojnkamjorn W. 2009, Thailand

yes

yes

yes

NR

NR

yes

yes

yes

yes

NR

yes

yes

yes

NR

fair

Piette J.D. 2011, USA

yes

yes

yes

NR

NR

yes

yes

yes

yes

NR

yes

yes

yes

yes

good

Cezaretto A 2012, Brazil

yes

yes

NR

NR

NR

yes

no

yes

yes

NR

yes

no

yes

NR

poor

Markle-Reid M. 2017, Canada

yes

yes

yes

no

yes

yes

yes

yes

yes

NR

yes

yes

yes

yes

good

Miklavcic J.J. 2020, Canada

yes

yes

yes

no

yes

no

yes

yes

yes

NR

yes

yes

yes

yes

good

Blackberry I.D. 2013, Australia

yes

yes

yes

no

yes

yes

yes

yes

yes

NR

yes

yes

yes

NR

good

van der Wulp I. 2012, Netherlands

yes

yes

yes

NR

NR

yes

yes

yes

yes

NR

yes

yes

yes

NR

fair

Davies M.J. 2008, UK

yes

yes

yes

NR

NR

yes

no

yes

yes

NR

yes

yes

yes

yes

fair

Du Pon E. 2019, Netherlands

yes

yes

yes

NR

NR

yes

yes

yes

yes

NR

yes

yes

yes

yes

good

Cortez D.N. 2017, Brazil

yes

yes

yes

NR

NR

yes

yes

yes

yes

NR

yes

yes

yes

NR

fair

Vadstrup E.S. 2011, Denmark

yes

yes

yes

no

yes

yes

yes

yes

yes

NR

yes

yes

yes

yes

good

Pauley T. 2016, Canada

yes

yes

yes

no

yes

yes

yes

yes

yes

NR

yes

NR

yes

yes

good

Siaw M.Y.L. 2017, Singapore

yes

yes

yes

NR

NR

yes

yes

no

no

NR

yes

yes

yes

yes

fair

Kulzer B. 2018, Germany

yes

yes

yes

NR

NR

yes

yes

yes

yes

NR

yes

yes

yes

yes

good

Browning C. 2016, China

yes

yes

yes

no

yes

yes

yes

yes

yes

NR

yes

yes

yes

NR

good

Kinmonth A.L. 1998, UK

yes

yes

yes

no

yes

yes

no

yes

yes

NR

yes

yes

yes

yes

good

Lamers F. 2011, Netherlands

yes

yes

yes

NR

yes

yes

yes

no

yes

NR

yes

yes

yes

yes

good

Abbreviation: NR, not reported.

Author-derived key for standardization: ≤ 5 POOR, 6-7 FAIR, ≥ 8 GOOD.

Author Response

We thank the reviewer for the valuable comments.

Reviewer 1

The authors could discuss more about the follow-up in the selected studies, in addition to the duration, similarity or differences in the results at the measurement or evaluation cut-off points of the patients, what trends were observed during those follow-ups and perhaps factors at which that these variations were attributed during the follow-ups. In table S2 in which the 13 quality criteria that give clarity to the results can be added.

We thank the reviewer for the valuable insights. We proceeded to discuss about the follow-up time and about the quality assessment results. Please, see lines 328-337.

Reviewer 2 Report

  1. need to include table for included studies before meta analysis
  2. how many studies were in Italian language ? and included in the final analysis 
  3. can add table for quality assessment 

Author Response

We thank the reviewer for the valuable comments.

Reviewer 2

Need to include table for included studies before meta analysis

Thank you for the suggestion. However, due to formatting restrictions, we put the table of the summary characteristics of the included studies in the supplementary file (Table S1).

How many studies were in Italian language ? and included in the final analysis

Among the included studies, there were not papers written in Italian.

Can add table for quality assessment

The table for quality assessment was put in a supplementary file (Table S2).

Reviewer 3 Report

Introduction: Some references are too old such as Wagnre 1998 , which should be converted to number! Please replace any references older than 2005 from Introduction and discussion…

Methods: The PROSPERO registration ID was not provided… The search strategies for all included databases should be provided in the supplementary materials...

Author Response

We thank the reviewer for the valuable comments.

Reviewer 3

Introduction: Some references are too old such as Wagnre 1998 , which should be converted to number! Please replace any references older than 2005 from Introduction and discussion…

We thank the reviewer for the valuable comment. We replaced most of the older references with newest ones. However, we cannot substitute the reference of Professor Edward Wagner since it developed the model we are referring to.

Methods: The PROSPERO registration ID was not provided… The search strategies for all included databases should be provided in the supplementary materials...

Thank you for the precious suggestion. We did not provide the PROSPERO registration ID since we did not register the systematic review in the PROSPERO database.

The search strategy for all the databases is now available in a new supplementary file.